# Effect of Astaxanthin on Tissue Transglutaminase and Cytoskeletal Protein Expression in Amyloid-Beta Stressed Olfactory Ensheathing Cells: Molecular and Delayed Luminescence Studies

**DOI:** 10.3390/antiox12030750

**Published:** 2023-03-19

**Authors:** Agatina Campisi, Giovanni Sposito, Rosaria Grasso, Julia Bisicchia, Michela Spatuzza, Giuseppina Raciti, Agata Scordino, Rosalia Pellitteri

**Affiliations:** 1Department of Drug and Health Sciences, University of Catania, 95125 Catania, Italy; 2CERNUT, Research Centre for Nutraceuticals and Health Products, University of Catania, 95125 Catania, Italy; 3Department of Physics and Astronomy “Ettore Majorana”, University of Catania, 95123 Catania, Italy; 4Institute for Biomedical Research and Innovation (IRIB), National Research Council, 95126 Catania, Italy; 5Laboratori Nazionali del Sud, National Institute for Nuclear Physics, 95123 Catania, Italy

**Keywords:** astaxanthin, tissue transglutaminase, olfactory ensheathing cells, amyloid-beta, self-renewal, delayed luminescence

## Abstract

Astaxanthin, a natural compound of *Haematococcus pluvialis*, possesses antioxidant, anti-inflammatory, anti-tumor and immunomodulatory activities. It also represents a potential therapeutic in Alzheimer’s disease (AD), that is related to oxidative stress and agglomeration of proteins such as amyloid-beta (Aβ). Aβ is a neurotoxic protein and a substrate of tissue transglutaminase (TG2), an ubiquitary protein involved in AD. Herein, the effect of astaxanthin pretreatment on olfactory ensheathing cells (OECs) exposed to Aβ(1–42) or by Aβ(25–35) or Aβ(35–25), and on TG2 expression were assessed. Vimentin, GFAP, nestin, cyclin D_1_ and caspase-3 were evaluated. ROS levels and the percentage of cell viability were also detected. In parallel, delayed luminescence (DL) was used to monitor mitochondrial status. ASTA reduced TG2, GFAP and vimentin overexpression, inhibiting cyclin D_1_ levels and apoptotic pathway activation which induced an increase in the nestin levels. In addition, significant changes in DL intensities were particularly observed in OECs exposed to Aβ toxic fragment (25–35), that completely disappear when OECs were pre-incubated in astaxantin. Therefore, we suggest that ASTA pre-treatment might represent an innovative mechanism to contrast TG2 overexpression in AD.

## 1. Introduction

Growing evidence highlights that oxidative damage to the brain represents an early event of neurodegenerative diseases, including Alzhèimer’s disease (AD), that is characterized by progressive cognitive impairment, behavior alterations and loss of memory. Oxidative stress, due to an imbalance between oxidants and antioxidants species, induces several changes in various pathways, leading to cellular control loss and cell death in AD.

It has been reported that dietary supplementation with antioxidants could play an important role in reducing oxidative stress, contributing to the onset responsible of AD progression. In particular, astaxanthin (ASTA; the Appendix A shows its structural conformation) a carotenoid from microalga *Haematococcus pluvialis* present in freshwater areas and produced by marine microorganisms, might represent a therapeutic potential for AD [1,2,3]. In AST is able to cross the blood–brain barrier (BBB), exerting positive effects on neurodegenerative diseases. ASTA might also prevent the formation of intracellular and extracellular protein aggregates formation induced by microtubule-associated protein tau and aberrant amyloid-beta (Aβ) products responsible for oxidative stress, neurotoxicity and inflammation, typical features of AD.

Previous reports have demonstrated that the treatment both in vitro and in vivo in AD experimental models with natural antioxidants was able to modulate the intracellular levels of tissue transglutaminase (TG2) and its isoforms [4,5,6]. TG2 is an ubiquitarian calcium-dependent protein responsible for cross-linking reactions, that lead to Aβ oligomerization and aggregation typical signs of AD [7]. Furthermore, TG2 possesses disulfide isomerase, GTPase and kinase activities [8]. Depending on intracellular localization, TG2 exerts different functions: when the protein is localized in the cytosol, it is involved in the apoptotic pathway and it possesses transamidating activity essential for its pro-apoptotic effect [9,10]. In contrast, TG2 localized in the nucleus shows kinase activity [11]. In addition, it shows two isoforms with different cellular functions: TG2-Long (TG2-L) and TG2-Short (TG2-S). In physiological conditions, TG2 presents a “closed conformation”, catalytically inactive, that is related to low intracellular Ca^2+^ levels accompanied by high GTP or GDP levels and promotes cell growth and survival. During pathological conditions, oxidative stress induced an enhancement of intracellular Ca^2+^ levels and a decrease of GTP/GDP binding to TG2, that shows “open conformation”, that it is catalytically active and involved in apoptosis and cell death. When the intracellular Ca^2+^ levels extend lengths of time, TG2 presents a “splice variant” lacking of carboxy-terminal portion that permits it to assume no “closed conformation”, inducing apoptosis [12]. In addition, findings indicated that TG2 plays a modulatory role in mitochondrial function in neurodegenerative diseases, including AD [13]. In particular, in AD, TG2 levels and its activity in the cerebral cortex are higher than those found in the controls and the protein colocalized with Aβ aggregates.

We previously demonstrated that olfactory ensheathing cells (OECs) peculiar olfactory glial cells used because of an early sign of neurodegeneration that is related with functionally reduced olfactory performance [14] exposed to full native peptide Aβ(1–42) and to its fragments Aβ(25–35) and Aβ(35–25), induced a significant increase of TG2 expression levels [4]. Furthermore, we showed that pre-treatment with Indicaxanthin, a natural antioxidant from *Opuntia ficus-indica*, L. Mill. [15,16] of Aβ exposed OECs, was able to reduce TG2 and its isoforms expression levels, counteracting the effect of Aβ [5]. In addition, it has been demonstrated that the delayed luminescence (DL), a phenomenon of photo-induced, ultra-weak and prolonged in time emission of optical photons, is able to verify the origin and possible changes attributed to the presence and alterations of ordered structures [17,18]. The investigations until now performed in in vivo and in vitro models, have shown DL emission with different intensities and different spectral components contribution, whose kinetics can be described as the sum of appropriately weighted compressed hyperbolas, typical of complex systems [19,20]. In particular, the experiments carried out on cell cultures of Jurkat-T cells [21,22], thyroid carcinoma [23] and glioblastoma [24], highlighted how DL emission changes its intensity and/or kinetic when treatments were able to activate apoptotic pathway and/or unbalance ROS levels and/or NAD^+^/NADH levels in mitochondria. A correlation with the structure of complex I of the mitochondrial respiratory chain (MRC) and with electron flow in MRC has been speculated [24,25].

In view of these considerations, we evaluated the effect of ASTA on TG2 localization and expression levels in OECs exposed to Aβ(1–42) and its fragments Aβ(25–35) and Aβ(35–25). Since AD is accompanied by gliosis [26], we also detected vimentin and glial fibrillary acid protein (GFAP) expression, used as a typical cytoskeletal marker. In addition, we assessed nestin, as cytoskeletal marker of neural precursors, and Cyclin D_1_ expression, a cell proliferation marker, that is co-expressed with nestin [5,27]. The percentage of cell viability, total reactive oxygen species (ROS) production and apoptotic pathway activation were also tested. To monitor the mitochondrial assessment in OECs exposed to Aβ and its fragments both with and without ASTA, DL emission intensity was also evaluated.

## 2. Materials and Methods

### 2.1. Materials

Aprotinin, leupeptin, Ethylenediaminetetraacetic Acid (EDTA) Phenylmethylsulfonyl Fluoride (PMSF), Ethylene Glycol-bis(β-aminoethyl ether)-*N*,*N*,*N*′,*N*′-Tetraacetic Acid (EGTA), Sodium Dodecyl Sulfate (SDS), cytosine arabinoside, phosphatase inhibitor cocktail II, full native peptide of Aβ(1–42), fragment Aβ(25–35), fragment Aβ(35–25), Dimethyl sulfoxide (DMSO), MTT Cell Viability Assay, Lab-Tek II Chamber-Slide Systems, paraformaldehyde and others materials were purchased from Sigma-Aldrich (Milan, Italy). Methanol and acetic acid were of LC grade and purchased from Merck (Milan, Italy). Trypsin, antibiotics, Fetal Bovine Serum (GIBCO), Normal Goat Serum (NGS, GIBCO), Phosphate Buffer Saline solution (PBS), Modified Eagle Medium (MEM) added with 2 mM GlutaMAX (GIBCO), Nitrocellulose Membrane Filter Paper Sandwich 0.45 µm pore size (Invitrogen), mouse monoclonal antibody against β-tubulin, anti-rabbit IgG horseradish peroxidase-conjugated and anti-mouse IgG horseradish peroxidase-conjugated were from Thermo Fisher Scientific (Milan, Italy). Bicinconinic acid method was bought from Pierce/Thermo-Scientific (Rockford, IL, USA). Mini-PROTEAN^®^ TGX™ Precast Protein Gels (4–15%), Mini Trans Blots Filter Paper, 10× Tris/Glycine/SDS buffer, 10× Tris/Glycine buffer, 4× Laemmli Sample Buffer, 2-mercaptoethanol and Precision Plus ProteinTM Standard Dual Color were from Bio-Rad Laboratories Srl (Milan, Italy). Mouse monoclonal antibody against TG2 (Neomarkers), mouse monoclonal antibody against nestin and cellular ROS/Superoxide Detection Assay were from Abcam (Milan, Italy). Rabbit monoclonal antibody against cyclin D_1_ was from Millipore (Milan, Italy). Mouse monoclonal antibody against GFAP and mouse monoclonal antibody against vimentin were from DAKO. Mouse monoclonal antibody against caspase-3 was from Becton-Dickinson (Milan, Italy). Cy3 goat anti-mouse and Fluorescein Isothiocyanate (FITC)-conjugated goat anti-mouse IgG antibody were from Jackson Immunological Research Laboratories Inc. (West Grove, PA, USA) Western Lightning Plus Enhanced Chemiluminescence Substrate was from Perkin-Helmer (Monza, Italy).

### 2.2. Methods

#### 2.2.1. OEC Cultures

OECs were obtained from 2-day-old mouse olfactory bulbs following experimental procedures on animal care according to the Italian law (no. 116/1992), the European Community Council Directive (86/609/EEC) and approved by the Ethical Committee (Catania University, Catania, Italy) and National Ministry of Health (permit number 174/2017-PR). Animals were kept in a controlled environment (23 ± 1 °C, 50 ± 5% humidity) with a 12 h light/dark cycle with food and water available *ad libitum*. The number of animals was reduced and their sufferance was minimal. 

Olfactory bulbs were removed from mice and processed in according with the method used by Pellitteri [28]. Tissue was digested with collagenase and trypsin. Successively, DMEM supplemented with 10% FBS (DMEM/FBS) was added in order to stop trypsization. Cells were plated in flasks and fed with complete DMEM/FBS. To reduce the number of dividing fibroblasts, an antimitotic agent, cytosine arabinoside (10^−5^ M), was added 24 h after initial plating; in addition, to reduce contaminating fibroblasts, OECs were transferred from one flask to a new one, following the method by Chuah and Au [29]. Confluent OECs were removed by trypsin and transferred in flasks and fed with DMEM/FBS at 37 °C in humidified 5% CO_2_–95% air mixture. Purified OECs were grown in DMEM/FBS both on glass coverslips and on 96 multiwells at a final density of 1 × 10^4^ cells/coverslip and on 25 cm^2^ flasks at a final density of 1 × 10^6^.

#### 2.2.2. Treatment of OECs

OEC cultures were separated in four different groups: (a) was considered as a control (CTR), both with the treatment and with the corresponding volume of DMSO (final concentration 0.02% *v*/*v*) used to solubilize full native peptide Aβ; (b) was exposed for 24 h to 10 μM Aβ(1–42) or Aβ(25–35) or Aβ(35–25), as reported in our previous work [4]; (c) in another group of OECs, ASTA (100 µM) was added in the medium for 24 h; (d) some OEC cultures were pre-treated with ASTA (100 µM) for 30 min and subsequently were exposed to 10 μM both of native peptide Aβ(1–42) and its fragments. Previous studies were performed in order to select the optimal concentration of astaxanthin through MTT. Cells were exposed at different concentrations of ASTA (50 and 100 μM) for 12, 24 and 48 h both in the absence of and in the presence of 10 μM Aβ(1–42) or Aβ(25–35) or Aβ(35–25). We chose, as the optimal concentration, 100 μM for 24 h. 

#### 2.2.3. MTT Test

For all different conditions, the viability of OECs was evaluated by a quantitative colorimetric method, 3-[4,5-dimethylthiazol-2-yl)-2,5-diphenyl] tetrazolium bromide (MTT) reduction assay [18]. Briefly, 1.0 mg/mL of MTT was added to each multiwell for 2 h at 37 °C. Successively, the supernatant was gently removed and MTT solvent (acid-isopropanol/SDS) was added, the cells were put on an orbital shaker for 10 min. The optical density was evaluated through a microplate spectrophotometer reader (Titertek Multiskan; Flow Laboratories, Helsinki, Finland) at λ = 570 nm in each well sample. Data were expressed as the percentage MTT reduction when compared with control cells.

#### 2.2.4. Total ROS

In all experimental conditions, the production of total ROS was tested through Cellular ROS Detection Assay, as suggested by the manufacturer’s instructions. The fluorescent products generated by the dye green for total intracellular ROS was visualized using a wide-field Zeiss fluorescent microscope (Zeiss, Germany) equipped with a standard green (λ_Ex_/λ_Em_ = 490/525 nm) filter set.

#### 2.2.5. Immunocytochemistry and CLSM Analysis

The expression of vimentin, glial fibrillary acid protein (GFAP), nestin, caspase-3 and TG2 in OECs was identified through immunocytochemical techniques. After 24 h, all cells were fixed through 4% paraformaldehyde in 0.1 M PBS for 30 min and then were incubated overnight at 4 °C in the following primary antibodies: mouse monoclonal antibody against GFAP (diluted 1:1000), mouse monoclonal antibody against vimentin (1:50), mouse monoclonal antibody against nestin (1:200), mouse monoclonal antibody against TG2 (1:200) and mouse monoclonal antibody against caspase-3 (1:500). Secondary antibodies used were: FITC anti-mouse (diluted 1:200) and Cy3 anti-mouse (diluted 1:500) for 1 h at room temperature and in dark conditions. Successively, immunostained coverslips were analysed on a Zeiss fluorescent microscope (Zeiss, Germany) and images were captured with an Axiovision Imaging System. The immunostained for TG2 was obtained using a Confocal Laser Scanning Microscope (CLSM) 510 Meta (Zeiss, Germany) and captured with an Axiovision Imaging System [4,30]. For the acquisition with CLSM, we used an Apo 63 X/1.4 oil immersion objective and the Argon (λ = 488 nm) and HeNe (λ = 543 nm) lasers. Images were acquired at the pixel resolution of 1024 × 1024 and processed in order to enhance brightness and contrast using the software ZEN2009 (version no 5.5.0.452).

In OECs where primary antibodies were omitted, no specific staining was observed.

#### 2.2.6. Isolation of Total Protein and Western Blot Analysis

All OEC cultures were collected in PBS, centrifugated and resuspended in lysis buffer composed of 150 mM NaCl, 50 mM Tris-HCl (pH 6.8), 1 mM EDTA, 0.1 mM PMSF, 10 µg/mL of aprotinin, pepstatin, leupeptin, incubated at 4 °C for 30 min, centrifuged at 12,000× *g* for 10 min at 4 °C and the supernatants containing total cell proteins were collected [5,18]. Extracted proteins were frozen at −80 °C, and the bicinchoninic acid method was used for quantitation. Total proteins (40 µg) were divided through 4–15% precast SDS–polyacrylamide gels and transferred to nitrocellulose membranes. Filters were then incubated with the following primary antibodies (diluted 1:1000): mouse monoclonal antibody against TG2 isoforms, cyclin D_1_ mouse monoclonal antibody against β-tubulin. Anti-rabbit and anti-mouse IgG horseradish peroxidase-conjugated were used as secondary antibodies. Western Lightning Plus-ECL Enhanced Chemiluminescence Substrate was used to employ each protein. Each blot was then scanned through ChemiDoc Imaging System (Bio-Rad, Milan, Italy) to visualize it. The integrated software was used to perform a densitometric analysis. Data obtained were normalized with β-tubulin.

#### 2.2.7. Delayed Luminescence Spectroscopy

DL from cell cultures was measured by using a suitable equipment operating in single photon counting mode [31]. Briefly, each sample was illuminated by a laser pulse (Laser Photonics LN 230 C) at wavelength λexc = 337 nm, pulse-width 0.6 ns and energy 13 ± 0.2 μJ/pulse. The DL emission in the wavelength range 350–850 nm was detected by a photomultiplier tube (Hamamatsu R7206-01 SEL), cooled at temperature −10 °C to reduce thermal noise, and recorded by using a Multi-Channel Scaler (ORTEC, Ametek, Berwyn, PA, USA) with 2 μs dwell time. To avoid photomultiplier blinding during illumination, an electronic shutter was used, imposing 10 μs time lag in acquisition after the laser pulse. To increase the DL signal, the same run was repeated 100 times. Spectral measurements were performed by using broadband interferential filters (Edmund Optics) at the central wavelengths 450 nm, 550 nm and 650 nm (50 nm FWHM), respectively. After treatments, the cells were detached by using trypsin/EDTA solution and incubated for 7 min at 37 °C. Then, 50% FBS was added in order to stop the trypsinization, and centrifuged at 200× *g* for 10 min. The cells were suspended in PBS, centrifuged again (200× *g* for 10 min) and resuspendend in PBS at the final cell density ≥10^6^ cell/mL. DL spectroscopy was performed on 20 μL of single drops at room temperature (21 ± 1 °C).

For each treatment, DL measurements have been performed in triplicate.

#### 2.2.8. Statistical Analysis

Statistical analysis was used to compare the data of different groups through a one-way analysis of variance (one-way ANOVA), followed by post hoc Holm–Sidak test. Data represent the mean ± S.D. of five separated experiments carried out in triplicate. Values were considered statistically significant with *** *p* < 0.0001, ^++^ *p* < 0.001, ^$$^ *p* < 0.001.

## 3. Results

### 3.1. Cell Viability

In order to evaluate the cell viability in all experimental conditions, the MTT test was performed. The optimal concentration of Aβ(1–42), Aβ(25–35) and Aβ(35–25) was 10 μM for 24 h, as reported in our previous studies [4]. Optimal ASTA concentration and time were chosen at 100 µM and 30 min, respectively. DMSO-treated OECs were used as controls, as no difference compared with OECs grown with only medium was detected. As showed by phase-contrast image (Figure 1A) and through MTT test (Figure 1B) a significant decrease in viability of OECs exposed to 10 μM Aβ(1–42) or Aβ(25–35) was found, while no effect on cell viability was evident in the reverse sequence of Aβ(35–25), when compared with controls. When OECs were pre-treated with 100 µM of ASTA and subsequently exposed to 10 µM Aβ(1–42) or Aβ(25–35) for 24 h, ASTA was able to prevent loss of cell viability as showed in the Figure 1A,B. In addition, in OECs pre-treated with ASTA and then stressed with Aβ(35–25) fragment no significant change was found.

### 3.2. Expression of Markers by Immunofluorescence

Different expression of markers through immunofluorescence was tested in OECs at each stage in controls and in culture exposed and not exposed to Aβ, with and without ASTA pre-treatment. Some proteins such as GFAP, a glial marker of growth, astrogliosis and differentiation; vimentin, a cytoskeleton marker and substrate of TG2; nestin, a marker expressed in stem cells; caspase-3 cleavage, an apoptotic marker and TG2 expression were investigated. No cell-specific staining was observed when all primary antibodies were omitted.

#### 3.2.1. GFAP and Vimentin Expression

GFAP and vimentin expression was evaluated in order to identify glial reactivity and differentiation in OECs exposed to Aβ(1–42) or to Aβ(25–35) or Aβ(35–25) fragments with and without ASTA pre-treatment. As showed in Figure 2 and Figure 3, Aβ(1–42) or Aβ(25–35) exposure to OECs induced a remarkable increase in GFAP and vimentin, respectively, with respect to controls and to cells treated with the fragment Aβ(35–35); both markers appeared more evident in OECs exposed to Aβ(25–35) with respect to Aβ(1–42) ones. OECs treated with ASTA did not show any modification in GFAP and vimentin expression when compared with controls (Figure 3). Those pre-treated with ASTA and subsequently Aβ(1–42) or Aβ(25–35) exposure induced a decrease of the GFAP and vimentin expression, making them similar to that detected in the controls. The results highlight that ASTA is able to counteract Aβ(1–42) and fragment Aβ(25–35) stress in OECs.

#### 3.2.2. TG2 Expression

Immunocytochemical procedures and CLSM analysis showed a different localization of TG2 positivity OECs depending on the growing conditions. A low staining for TG2 was found in the cytosol of control cells, while an intense staining was revealed in the OEC cytosol in the Aβ(1–42) and Aβ(25–35)-treated, even if more marked in Aβ(25–35)-treated cells. When the OECs were exposed to Aβ(35–25), a light increase for TG2 cell positivity was revealed to be prevalently localized in the cytosol. In 100 µM ASTA-treated cells, TG2 staining was similar to the controls. In the cells exposed to Aβ(1–42) and Aβ(25–35), ASTA provoked a decrease on the TG2-positive cell number. When the OECs were treated with ASTA and exposed to Aβ(1–42), TG2 was localized into the nucleus, while it was prevalently localized in the cytosol when exposed to Aβ(25–35) and pre-treated with ASTA (Figure 4). In the ASTA pre-treated OECs that were then exposed to the fragment Aβ(35–25), a weak staining for TG2 was found.

#### 3.2.3. Nestin Expression

As shown in Figure 5, we found a significant reduction of nestin-positive cells when exposed to Aβ(1–42) and Aβ(25–35), while in Aβ(35–25)-exposed cells, nestin immunolabeling was increased. The pre-treatment of ASTA (100 µM) for 24 h did not induce any changes related to the number of nestin positivity, which was very similar to the levels observed in the controls. A considerable increase of OECs pre-treated with ASTA and subsequently exposed to Aβ(1–42) or Aβ(25–35) was reported compared with Aβ(1–42) and Aβ(25–35)-exposed cells (Figure 4). A light nestin positivity in ASTA pre-treated cells and then stressed to Aβ(35–35) was detected.

This result highlights that ASTA pre-treatment is able to stimulate TG2 repair activity in OECs when exposed to Aβ, activating also the stem self-renewal through the augment of the positivity for nestin.

#### 3.2.4. Caspase-3 Cleavage Immunolabeling

We evaluated the caspase-3 cleavage immunocytochemically to verify the TG2-mediated apoptotic pathway in OECs stressed to Aβ(1–42) or Aβ(25–35). The results obtained demonstrated that cell positivity for caspase-3 was almost absent in PBS, DMSO and Aβ(35–25)-treated cells; however, on the contrary, in cell Aβ(1–42) or Aβ(25–35) exposed, an increased activation of their positivity for caspase-3 with a major distribution into cytoplasm, and an increase in cell size was detected. In addition, in OEC pre-treated with 100 µM ASTA, no cell resulted as immunolabeled with caspase-3. When OECs were ASTA pre-treated and subsequently exposed to Aβ(1–42) or Aβ(25–35), the caspase-3 positivity was very reduced reaching expression levels similar to the controls (Figure 6). No positivity for caspase-3 was detected in OECs pre-treated with ASTA and then exposed to the fragment Aβ(35–25). These results highlight how TG2 is able to control the activation of apoptotic pathway in both Aβ-exposed OECs and those pre-treated with ASTA.

### 3.3. Total ROS Generation

The staining of total intracellular ROS levels was performed to monitor the intracellular oxidative status, in OECs pre-treated and untreated with ASTA exposed for 24 h to Aβ(1–42) or Aβ(25–35) or to Aβ (35–25) as showed in the Figure 7. Aβ exposure caused a significant increase in total ROS levels, when compared with the controls. ASTA pre-treatment caused a strong decrease of total intracellular ROS levels production in OECs exposed to Aβ(1–42) or to Aβ(25–35), when compared with the controls and Aβ(35–25). These results demonstrated that Aβ(1–42) and Aβ(25–35) increased the levels of total ROS and that ASTA pre-treatment counteracted the modification of the oxidative status modified by Aβ to control values.

### 3.4. TG2 Expression Levels Immunolabeling

To assess the effect of ASTA pre-treatment on OECs exposed to Aβ(1–42) or Aβ(25–35) or Aβ(35–25) on the role played by TG2, we have evaluated its expression levels, through Western blot and their relative densitometric analysis (Figure 8A,B). In cells treated with PBS and DMSO, used as control, and in OECs exposed to 100 μM of ASTA, TG2 was expressed at low levels. Aβ(1–42) and Aβ(25–35) treatments induced a significant increase of protein expression levels when compared with PBS and DMSO. In cultures exposed to Aβ(35–25), no significant modification in TG2 expression was detected. ASTA pre-treatment in OECs exposed to Aβ(1–42) caused a TG2 significant decrease, in comparison to cells treated with Aβ(1–42) alone. Differently, the pre-treatment with ASTA in Aβ(25–35)-exposed OECs showed a slight increase of TG2 expression, when compared with controls (PBS and DMSO). In addition, ASTA pre-treatment provoked a significant decrease of the TG2 expression when compared with the controls, with ASTA alone and with Aβ(35–25). All results were confirmed, after normalization with β-tubulin, through densitometric analysis performed in each condition. Our results showed that Aβ treatment both with and without ASTA differently modulates TG2 acting on apoptotic pathway activation and cell self-renewal capacity.

### 3.5. Cyclin D_1_ Expression Levels

Cyclin D_1_ expression was examined to verify the TG2 role played in the cellular repair induced by ASTA on Aβ exposure in OECs.

Figure 9A,B represents the Western blot and densitometric analysis for cyclin D_1_ on OECs exposed to Aβ(1–42), Aβ(25–35) and Aβ(35–25) in the absence and in the presence of 100 μM ASTA. In OECs exposed to Aβ(1–42) peptide, we found a significant decrease in cyclin D_1_ expression levels, when compared with the controls (PBS and DMSO). Furthermore, the treatment of cells with the toxic Aβ(25–35) induced a very strong reduction of cyclin D_1_ expression compared with Aβ(1–42) peptide, PBS and DMSO; while the treatment of OECs with no-toxic fragment Aβ(35–25) did not cause important modifications in cyclin D_1_ expression levels. When cells were pre-treated with ASTA, there was no significant change of cyclin D_1_ expression levels. The pre-treatment of cells with ASTA and following exposed to Aβ(1–42), highlighted an important increase of the cyclin D_1_ expression level when compared with Aβ(1–42) treated ones and controls. In addition, we observed a significant increase of cyclin D_1_ expression levels in OECs pre-treated with ASTA and Aβ(25–35), as compared with cells treated with Aβ(25–35) alone. 

### 3.6. DL Spectroscopy

Spectroscopic studies of DL were conducted on untreated OEC cultures, used as controls, and on the OEC cultures exposed to 10µM of native Aβ peptide (1–42) or toxic Aβ fragments (25–35) or (35–25) for 24 h, respectively. The effects of pre-incubation for 30 min in 100 µM of ASTA were also evaluated in all experimental conditions. DL emission was acquired in the time interval between 10 μs and 100 ms after the laser pulse was turned off, both in the visible spectrum, 350 ÷ 850 nm (λ_vis_), and in the intervals 425 ÷ 475 nm (λ_450nm_), 525 ÷ 575 nm (λ_550nm_) and 625 ÷ 675 nm (λ_650nm_) in which mitochondrial natural fluorescent molecules emit. Figure 10 shows DL time trend decays, that is DL intensity I(t) as function of discrete-time t, in the visible spectrum. It is possible to observe multimodal behaviours in DL kinetics as a function of the treatments.

The figure shows that the treatment of OECs with DMSO did not alter the decay kinetics of the DL emission, when compared with the controls. OECs exposure to Aβ(1–42) and its fragments altered the decay kinetics as a function of the used treatment. More precisely, OEC exposure to Aβ(1–42) gave rise to a small change at long DL times, while the use of the fragment, especially the toxic fragment Aβ(25–35), seemed to significantly change the decay dynamics both in intermediate and long times. To compare the effects induced by different treatments on the cells, DL integral emission, DLI, that is the total number of photons emitted (DLI $=\int_{t_{0}}^{t_{f}}I\left(t\right)dt$), were evaluated. The measurements were performed in different days. Thus, in order to remove the dependence on cell density, the total counts from each sample (DLI_s_) were normalized to the total counts from untreated OECs prepared at the same day and assumed as control (DLI_c_). Average values of such normalized total emission, along with the corresponding standard error, are reported in Figure 11. No effect on DL total emission was observed both in the controls and in the native peptide stressed cells, while DL total emission was significantly affected by the exposition to the fragments, especially to the toxic fragment Aβ(25–35).

Figure 11 also shows the effect of pre-incubation for 30 min in 100 µM of ASTA. A drastic reduction in the total emission from samples exposed to Aβ(25–35) and Aβ(35–25) fragments, after ASTA pre-incubation, to values comparable with the control can be observed. To get more insight in the phenomenon, a spectral analysis of DL emission was performed on sample exposed to Aβ peptide and fragments, before and after pre-incubation with ASTA (100 µM, 30 min). As above said, the three spectral band centered at λ_em_ 450 nm, 550 nm and 650 nm were considered. Actually, these three components represent about 80% of the entire emission spectrum 350 ÷ 850 nm that can be revealed by the experimental apparatus. As reported in the Figure 12, no significant changes in the DL spectrum, in DMSO and 10 μM Aβ(1–42)-treated cells were observed when compared with the controls. In contrast, a significant decrease in the blue component (λ_em_ = 450 nm) and an increase in the green/yellow (λ_em_ = 550 nm) and no significant change in red (λ_em_ = 650 nm) components are evident when OECs were exposed to Aβ(25–35) and Aβ (35–25) fragments, the effect being more pronounced in presence of the toxic Aβ(25–35) fragment, when compared with the DMSO and Aβ(1–42)-treated OECs (Figure 12A). Such differences almost disappeared in ASTA-treated cells, as shown in Figure 12B, even if the red component (λ_em_ = 650 nm) of samples exposed to the toxic Aβ(25–35) fragment remains greater than the control one.

## 4. Discussion

Several studies have shown that the oxidative damage in the brain of patients with AD is correlated with the aberrant formation of Aβ [32], that represents the prevalently component of senile plaques, neutrophil threads and neurofibrillary tangles [33]. Aβ is also responsible of mitochondrial disfunctions and tau hyperphosphorylation contributing to the pathology [34].

It has been reported that ASTA, an antioxidant, anti-apoptotic and anti-inflammatory compound, has stronger neuroprotective effects in neurological diseases, including AD, compared with other carotenoids, such as lutein, β-carotene, canthaxanthin, zeaxanthin [35], counteracting oxidative stress and inhibiting apoptotic cell death [36]. In addition, it plays a key role in the pathway of the PI3K/Akt down-regulating caspase-3 activation [37].

This research highlights the effect of ASTA on localization and expression levels of TG2 in OECs exposed to Aβ(1–42) or its fragments. Since GFAP and vimentin, cytoskeleton proteins, which are involved in AD and show an important role in astrogliosis, were tested [38]. In addition, we assessed the effect played by ASTA on cellular repair and self-renewal evaluating cyclin D_1_ and nestin expression [27,39]. Total ROS production and the percentage of cellular viability were also tested in the same experimental conditions. To monitor the mitochondrial assessment in Aβ-stressed OECs both in the absence and in the presence of ASTA, DL emission intensity was also evaluated.

CLSM analysis showed that the TG2 was differently localized in OECs depending on treatment type. In Aβ(1–42) and in Aβ(25–35)-treated cells, TG2 was prevalently localized in the cytosol. This observation demonstrated that Aβ exposure, increasing intracellular Ca^2+^ levels, induced the pro-apoptotic role exerted by TG2. In fact, in these conditions, we found a significant increase of the apoptotic pathway as revealed by the caspase-3 detection, as showed in the Figure 6. In OECs exposed to Aβ(1-42), ASTA pre-treatment induced the localization of TG2 prevalently in the nucleus. This effect might be due to the reparative role played by nuclear TG2 stimulated by ASTA pre-treatment. In fact, ASTA is able improved stem cell potency increasing neural proliferation, as confirmed by the increase of cyclin D_1_ and nestin levels found in these experimental conditions, when OECs were exposed with the Aβ(1–42) alone. In contrast, in ASTA pre-treated cells and those exposed to Aβ(25–35), TG2 appears localized both in the cytosol and in the nucleus. This result could be related to the role played by splice variant of TG2 (TG2-S) that is related to an excessive enhancement of intracellular Ca^2+^ levels induced by the toxic Aβ(25–35) fragment. This observation is supported by the low caspase-3 cleavage positivity in OECs pre-treated with ASTA subsequently exposed to Aβ(25–35). In parallel, ASTA pre-treatment reduced the expression of GFAP and vimentin in OECs exposed to Aβ subsequently, demonstrating that it was able to reduce the gliosis induced by Aβ. This data is in agreement with other observations showing that ASTA stimulates the expression of proteins involved in brain repair, such as GFAP, promoting communication and nervous cell regeneration [40].

It was observed that ASTA, for its antioxidant ability, is able to re-establish the oxidative status modified by Aβ exposure in OECs to the control levels. Recent experimental work investigated the relationship between DL and mitochondrial status in Saccharomyces cerevisiae [41], while the effects of different agents with mitochondrial target on DL from different cells were also evaluated [21,22,23,24,25]. In the present study, the presence of Aβ peptide, especially the toxic fragment, affects DL response. As showed in the Figure 11, an increase (more than twice) in the DL emission yield on OECs exposed to the Aβ(25–35) fragment when compared with the control and with the native peptide Aβ(1–42). This effect might be related to the accumulation of extracellular Aβ, that is responsible for cell cycle alterations and oxidative stress [42]. In particular, it has been demonstrated in neuron cultures that Aβ(25–35) fragment, while mimicking the toxicological and aggregating properties of the native peptide Aβ(1–42), is more toxic and causes greater oxidative stress at the level of membrane proteins, and the formation of Aβ aggregates is faster than the native peptide Aβ(1–42). In this regard, it has been investigated that the oxidative stress induced by the toxic fragment of Aβ is different with respect to those caused by Aβ(1–42), also elucidating the role of the C-terminal methionine Met-35 in the toxic properties of Aβ(25-35) [43]. Interestingly, Varadarajan and coworkers revealed a robust increase (2.5 times control value) in membrane protein oxidation due to the toxic fragment Aβ(25–35), by evaluating protein carbonyl levels. We found that DL emissions showed a high sensitivity to the effect of the toxic fragment Aβ(25–35) when compared with the other Aβ (Figure 11); therefore, we can assess that its effect on DL emission could be related to the different mechanisms that govern their toxicological properties.

Spectral emission showed that the most significant differences were observed in OECs exposed to the toxic fragment Aβ(25–35), as reported in Figure 12A. This result can be related to a decrease of reduced forms of nicotinamide adenine dinucleotide (NADH) consumption/production and an increase of ROS production. Indeed, a similar spectral emission with an increase of the green/yellow component, of the ultraweak photo-induced delayed photoemission, when compared with the control, was observed in Jurkat cell line treated with Rotenone, a familiar specific inhibitor of mitochondrial respiration chain complex I with anticancer activity attributed to the induction of apoptosis [25]. In addition, the same researchers considered flavin mononucleotide (FMN) as mostly involved in green/yellow emission. On the other hand NAD(P)H and flavins are the main responsible of the overall autofluorescence signal from single cells, being strictly involved in reaction pathways of energy metabolism [44]. Indeed, it has been observed that the Aβ(25–35) fragment induced mitochondrial damage that leads to cell death through the activation of the apoptotic pathway, inhibiting mitochondrial respiration and inducting mitochondrial swelling, for the opening of the mitochondrial permeability transition pore resulting in a loss of the mitochondrial transmembrane potential [45].

It is worth noting that ASTA pre-incubation of OECs induced a recovery in DL intensities, both in the total and in the spectral components, versus control ones (Figure 11 and Figure 12B). This result, supported by biochemical tests, highlighted that in OECs exposed to Aβ(25–35) after pre-incubation with ASTA, the red component was reduced even if it remained greater than the control one. In fact, the ROS level was reduced, but remained greater when compared with the control value (Figure 7).

## 5. Conclusions

Taken together, our results highlight that ASTA in OECs exert a protective effect against Aβ toxicity, and that they modulate the aberrant TG2 expression up-regulated by the Aβ exposure of the cells. OECs, as cells capable of expressing and releasing neurotrophic factors, might represent a promising tool for the regeneration of neural tissue of AD.

## Figures and Tables

**Figure 1 antioxidants-12-00750-f001:**
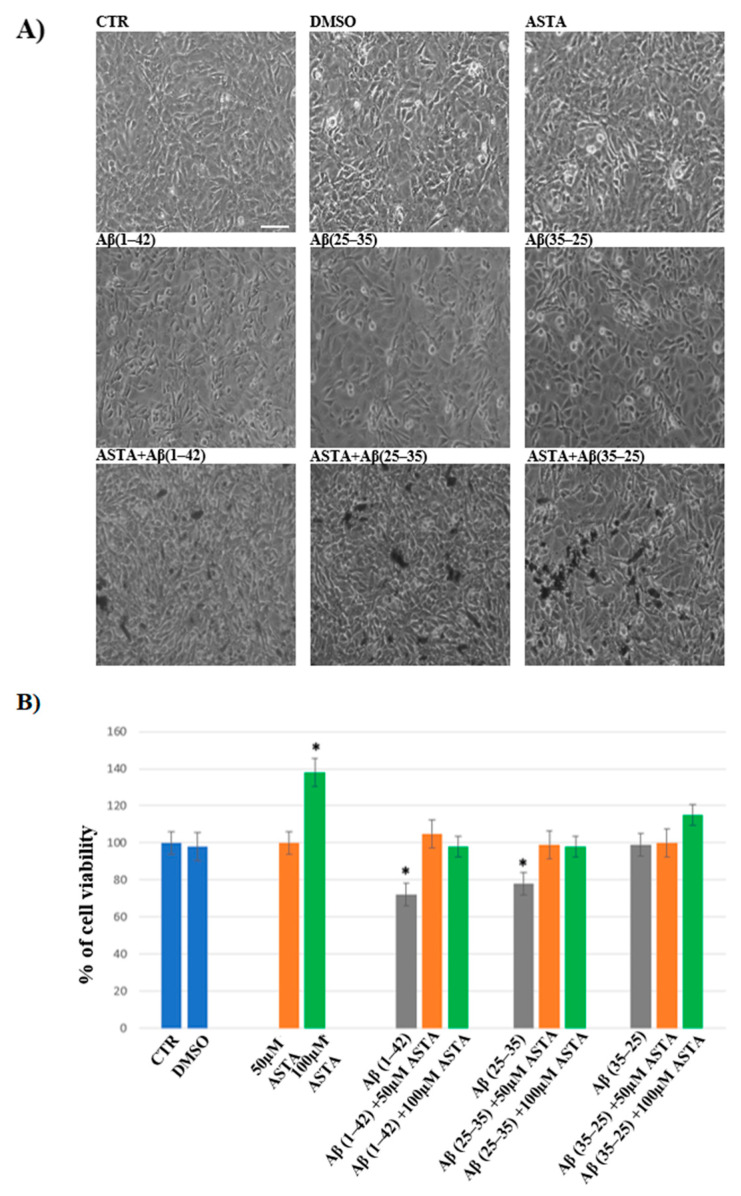
Phase-contrast images (**A**) and percentage of cell viability (**B**) in OECs performed through MTT test. OECs were exposed to Aβ(1–42) or Aβ(25–35) or Aβ(35–25) for 24 h; OECs pre-treated with ASTA for 30 min and exposed to Aβ(1–42) or Aβ(25–35) or Aβ(35–25) for 24 h. Data were represent the mean ± S.D. of five separated experiments in triplicate. * *p* < 0.05 significant differences vs. CTR. Scale bars 50 µm.

**Figure 2 antioxidants-12-00750-f002:**
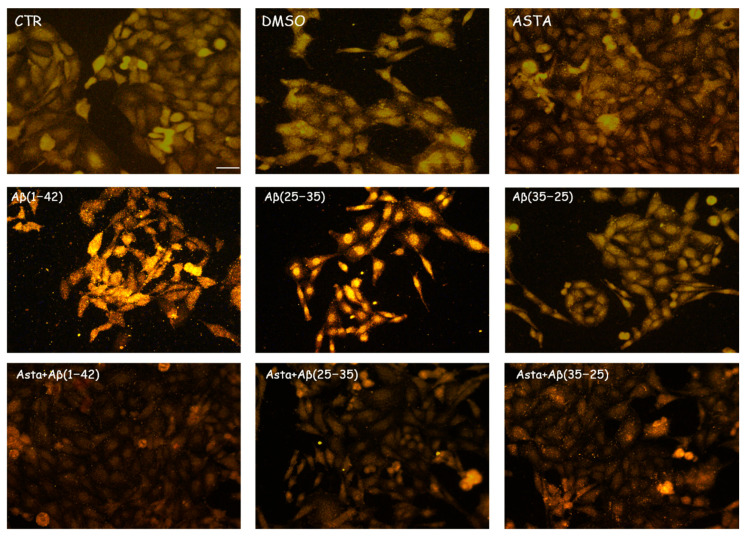
Immunocytochemistry for GFAP in OECs. Images show the ASTA effect in OEC treated to different conditions. The treatment of cultures with ASTA did not induce changes for the cells positivity for GFAP. The pre-treatment with (ASTA) restored the levels of the protein to control values, modified by the exposure to Aβ(1–42) or Aβ(25–35). Scale bar 20 µm.

**Figure 3 antioxidants-12-00750-f003:**
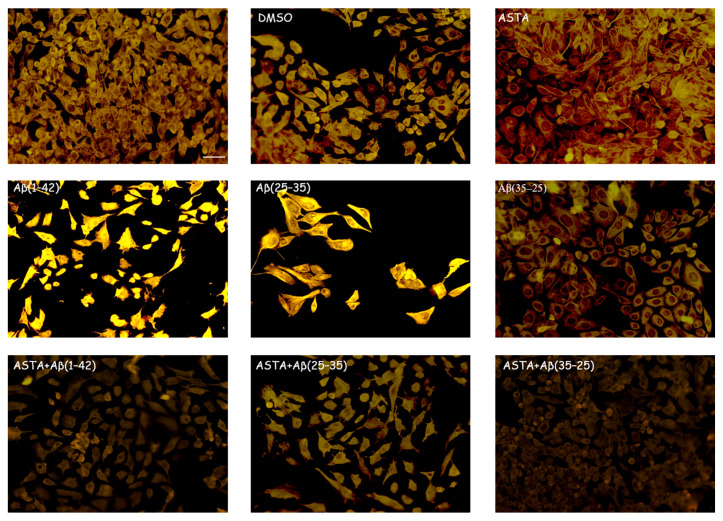
Immunocytochemistry for vimentin. Images show ASTA effect in OEC treated to different conditions. ASTA pre-treatment in cultures did not induce changes for the positivity of the cells for vimentin. The pre-treatment with ASTA restored the levels of the protein to control values, modified by Aβ(1–42) or Aβ(25–35) exposure. Scale bar 20 µm.

**Figure 4 antioxidants-12-00750-f004:**
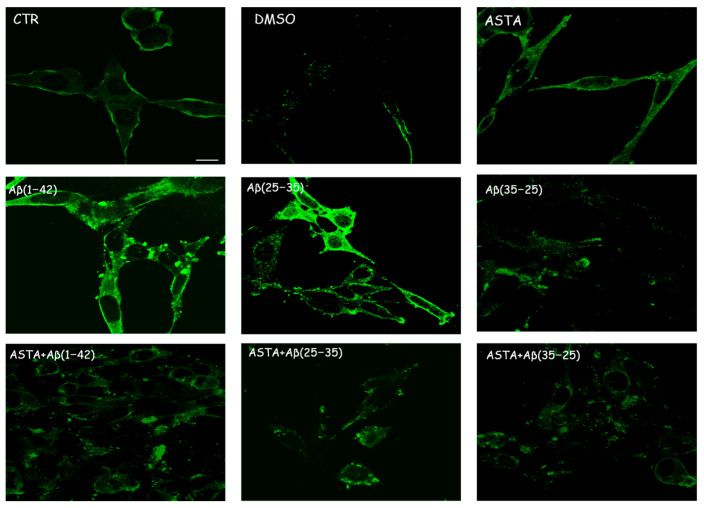
CLSM analysis of labeling immunocytochemistry for TG2 in OECs. Images show a different protein localization and positivity depending on the growing conditions. Scale bar 20 µm.

**Figure 5 antioxidants-12-00750-f005:**
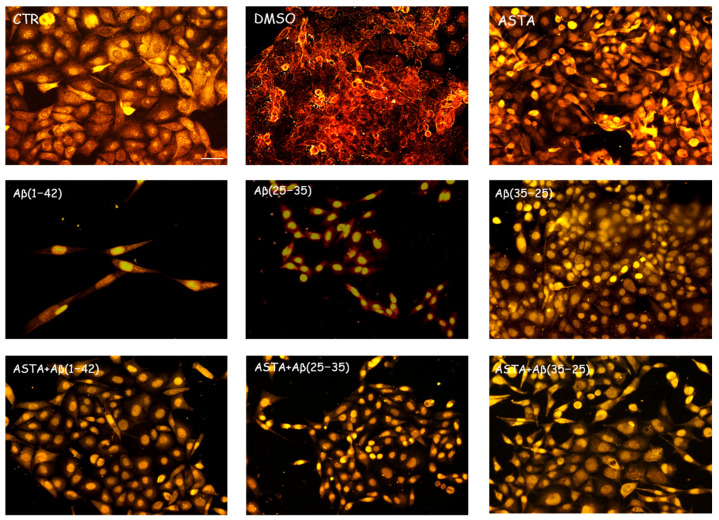
Immunocytochemistry for nestin in OECs. Images show ASTA effect in OECs treated to different conditions. The pre-treatment with 100 µM ASTA for 24 h restored the levels of the protein to control values, modified by the exposure to 10 μM Aβ(1–42) or Aβ(25–35). Scale bar 20 µm.

**Figure 6 antioxidants-12-00750-f006:**
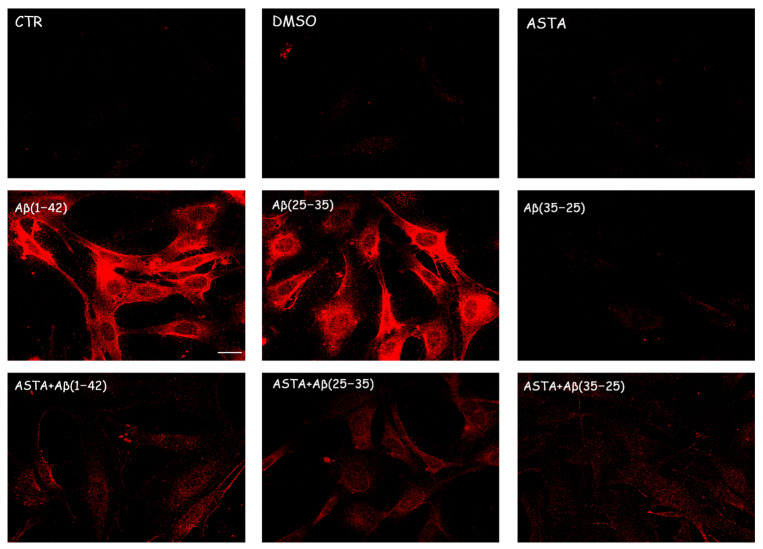
Immunocytochemistry for caspase-3 in OECs exposed to different conditions. Effect of 100 µM ASTA in Aβ(1–42) or Aβ(25–35) or Aβ(35–25)-stressed cells for 24 h. Scale bar 20 µm.

**Figure 7 antioxidants-12-00750-f007:**
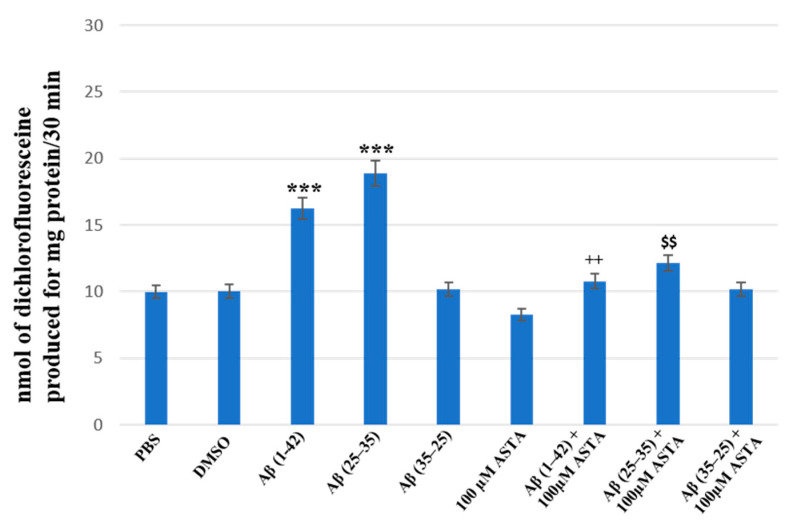
Total intracellular ROS levels generation in OECs in different conditions: PBS, DMSO, 100 µM ASTA, 10 μM Aβ(1–42) or Aβ(25–35) or Aβ(35–25) both in the absence and in the presence of 100 µM ASTA for 24 h. *** *p* < 0.0001 vs. CTR (PBS and DMSO); ^++^ *p* < 0.001 vs. Aβ(1–42); ^$$^ *p* < 0.001 vs. Aβ(25–35).

**Figure 8 antioxidants-12-00750-f008:**
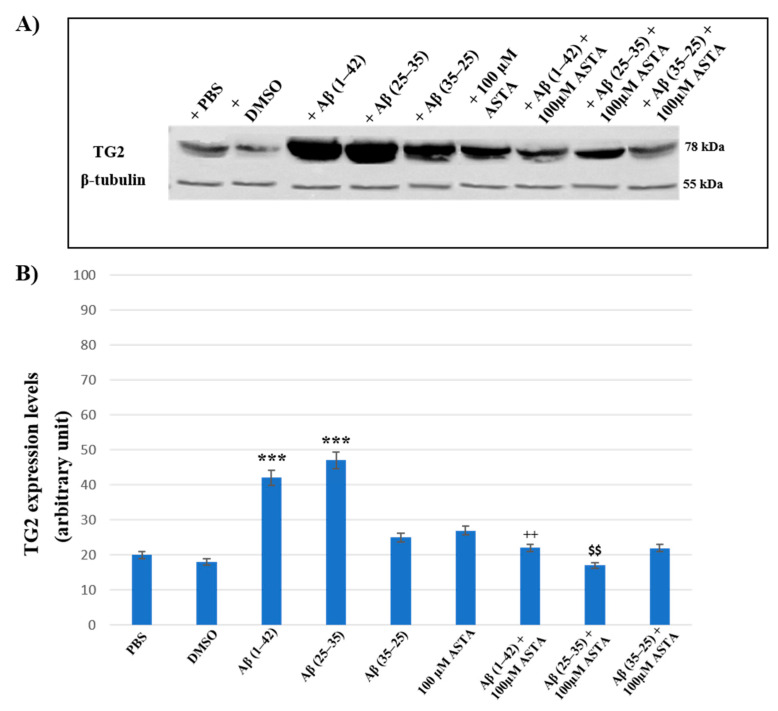
(**A**) Immunoblots obtained through Western blotting analysis for TG2 expression in total cellular lysate from OECs in different conditions: PBS, DMSO, 100 µM ASTA, 10 μM Aβ(1–42) or Aβ(25–35) or Aβ(35–25) both in the absence and in the presence of 100 µM ASTA for 24 h; (**B**) densitometric analysis of TG2 levels performed after normalization with β-Tubulin. The results are expressed as the mean ± S.D. of the values of five separate experiments performed in triplicate. *** *p* < 0.0001 vs. CTR (PBS and DMSO); ^++^ *p* < 0.001 vs. Aβ(1–42); ^$$^ *p* < 0.001 vs. Aβ(25–35).

**Figure 9 antioxidants-12-00750-f009:**
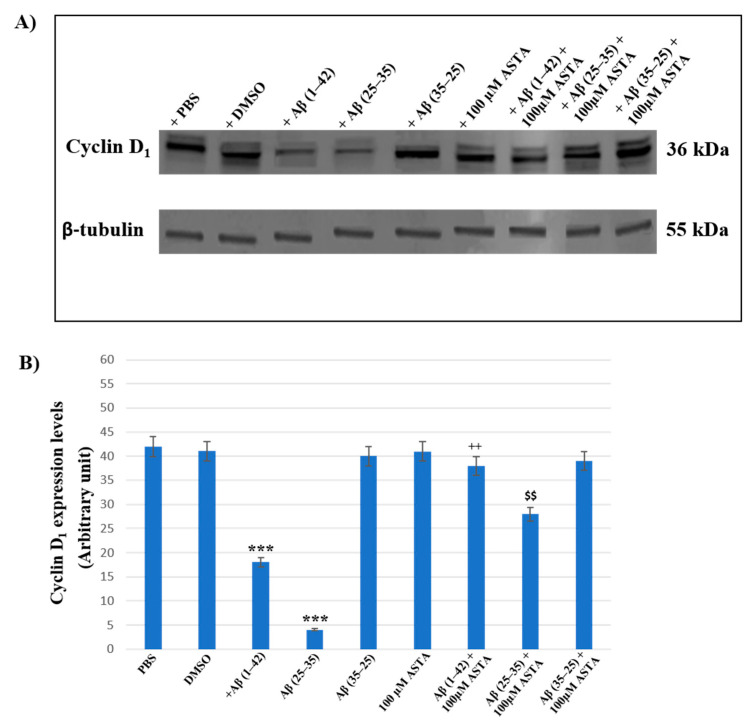
Western blotting analysis. (**A**) Representative immunoblot through Western blotting analysis for cyclin D_1_ expression levels in total cellular lysates from OECs in different conditions: control, DMSO, 100 µM ASTA, 10 μM Aβ(1–42) or Aβ(25–35) or Aβ(35–25) both in the absence and in the presence of 100 µM ASTA for 24 h. (**B**) Densitometric analysis of cyclin D_1_ expression levels carried out after normalization with β-tubulin. The results are expressed as the mean ± S.D. of the values of five separate experiments performed in triplicate. *** *p* < 0.0001 vs. CTR (PBS and DMSO); ^++^ *p* < 0.001 vs. Aβ(1–42); ^$$^ *p* < 0.001 vs. Aβ(25–35).

**Figure 10 antioxidants-12-00750-f010:**
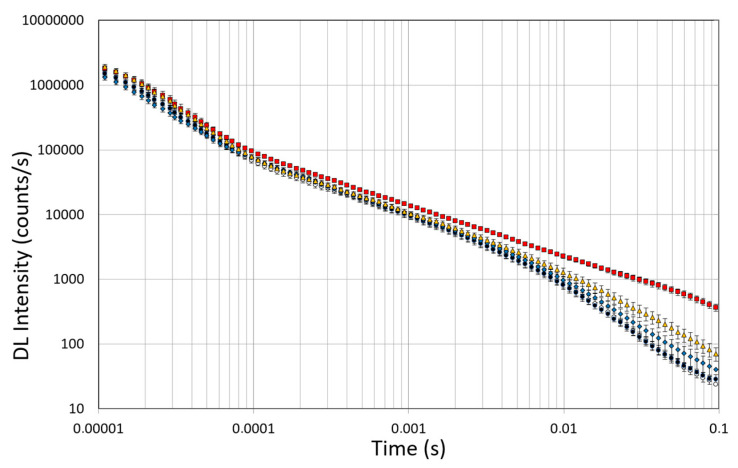
Time trend of DL emitted by OECs cultures in visible range (350 ÷ 850 nm). (Open circle) control cells; (grey circle) OECs exposed to DMSO, (diamond) OECs exposed to Aβ(1–42), (square) OECs exposed to Aβ(25–35); (triangle) OECs exposed to Aβ(35–25). Results are expressed as the mean ± S.E. of the values of at least three biological replicates in triplicate.

**Figure 11 antioxidants-12-00750-f011:**
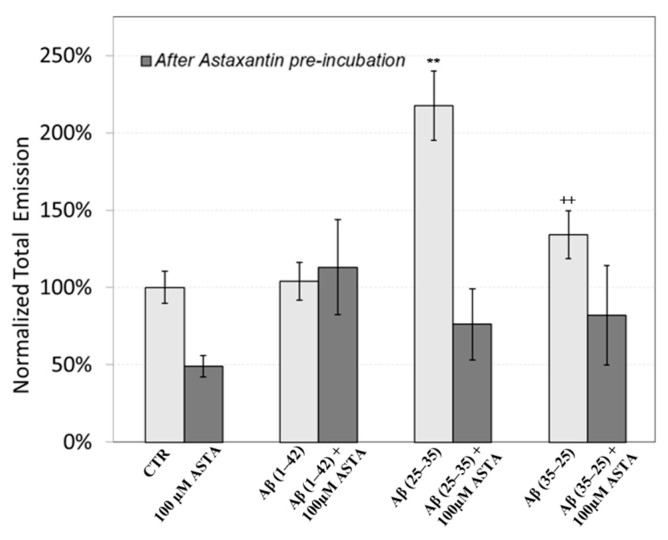
DL integral emission of OECs exposed to: 100 µM ASTA, 10 μM Aβ(1–42) or Aβ(25–35) or Aβ(35–25) both in the absence and in the presence of 100 µM ASTA for 24 h normalized to DL integral emission in control cell cultures in the visible range (350 nm ÷ 850 nm). Results are expressed as the mean ± S.E. of the values of at least three biological replicates in triplicate. Before (light grey) and after (grey bar) pre-incubation with ASTA (100 µM, 30 min). ** *p* < 0.001 vs. CTR (PBS and DMSO); ^++^ *p* < 0.001 vs. CTR (PBS and DMSO).

**Figure 12 antioxidants-12-00750-f012:**
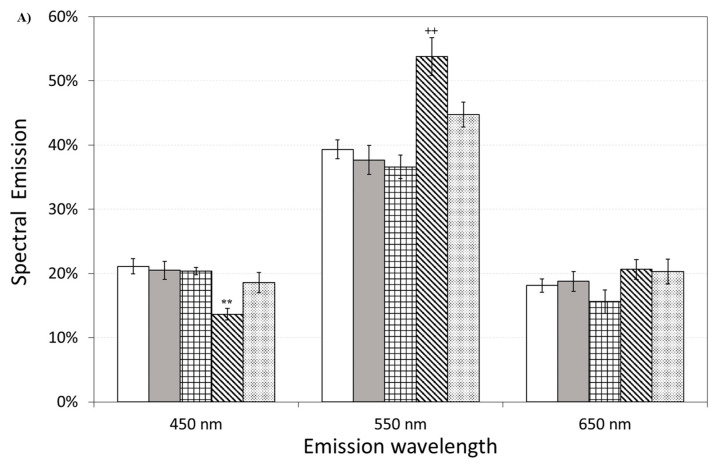
Delayed luminescence emission spectra, before (**A**) and after (**B**) pre-incubation with ASTA (100 µM, 30 min). (**A**) (white bar) untreated OECs (CTR); (dark grey bar) OECs exposed to DMSO; (squared bar) OECs exposed to Aβ(1–42); (hatched bar) OECs exposed to Aβ(25–35); (dotted bar) OECs exposed to Aβ(35–25). ** *p* < 0.001 and ^++^ *p* < 0.001 vs. CTR. (**B**) (white bar) untreated OECs (CTR), (dark grey bar) OECs pre-incubated with ASTA; (squared bar) OECs pre-incubated with ASTA and exposed to Aβ(1–42); (hatched bar) OECs with ASTA and exposed to Aβ(25–35); (dotted bar) OECs pre-incubated with ASTA and exposed to Aβ(35–25). Results are expressed as the mean ± S.E. of the values of at least three biological replicates in triplicate.

## Data Availability

The data presented in this study are available in the article.

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
