# Peer review of "Effect of Astaxanthin on Tissue Transglutaminase and Cytoskeletal Protein Expression in Amyloid-Beta Stressed Olfactory Ensheathing Cells: Molecular and Delayed Luminescence Studies"

_antioxidants, 2023, doi:10.3390/antiox12030750_

Round 1

Reviewer 1 Report

The manuscript presents the effect of astaxanthin on the expression level of Aβ(142) or by Aβ(2535) or Aβ(3525), and TG2 21, ROS level, and delayed luminescence of olfactory ensheathing cells, proving the protective role of the antioxidant.

Remarks:

What was the purity of the cell population used (% of ORCs)?

The text requires linguistic check and amendment. Some but by far not all points are listed below.

Lines 154/155: “to select the optimal concentration of Astaxanthin through MTT”, the phrase may be not clear. I understand that a concentration high enough to exhibit an effect but not causing a significant loss of cell viability  (estimated with MTT) was selected. Could the authors present more explicitly the reason (and perhaps data) for selecting the ASTA concentration chosen?

Line 28: “almost of DL intensity”, please reformulate

Line 43: “presents”, please change to “present”

Lines 45/46: “ASTA might also prevent the intracellular and extracellular protein aggregates”, perhaps better: “… formation of…”

Line 47: ” responsible of”, rather: “responsible for”

Line 57: “TG2 localization in the nucleus plays kinase activity”, please reformulate; localization cannot play activity; perhaps “TG2 localized… shows kinase activity”?

Line 74: “[15,16] on”, rather: ” [15,16] of”

Line 113: “bought by”, rather: “bought from”

Line 131: Can the authors provide the permit number?

Line 193: omitted or subtracted?

Line 237: “performed”, better “assay … was performed” or “viability… was evaluated”

Line 240: ” were choose”, should be “were chosen”

Line 247: is “re-establish” the right term? ASTA rather prevented loss of viability than re-established it (I do not believe in cell resurrection)

Figure 5. “Immunocytochemistry for anti-Nestin” is not an optimal legend. The authors studied nestin with antinestin. The same refers to Figure 6.

Line 346: ” ASTA pre-treatment counteracted the oxidative status”, please re-formulate; perhaps “counteracted the modification of the oxidative status…”

Figure 8A: Is a representative blot chosen for the Figure? It seems that the effect of DMSO was stronger on the blot than represented by the mean value on Figure 8B.

Line 523: “C-terminus of methionine Met-35”, not clear; “C-terminal methionine”?

Author Response

Lines 154/155: “to select the optimal concentration of Astaxanthin through MTT”, the phrase may be not clear. I understand that a concentration high enough to exhibit an effect but not causing a significant loss of cell viability (estimated with MTT) was selected. Could the authors present more explicitly the reason (and perhaps data) for selecting the ASTA concentration chosen?

Answer: We used MTT test to assess the percentage of cell viability, in order to establish the better concentration of Astaxanthin, as well known for its anti-inflammatory, anti-oxidant and anti-apoptotic activities, in addition it is able to cross the blood-brain barrier (Liu X, Osawa T. Astaxanthin protects neuronal cells against oxidative damage and is a potent candidate for brain food. Forum Nutr. 2009;61:129-135. doi: 10.1159/000212745). Our results highlighted that the pretreatment of OECs with Astaxanthin was able to prevent cell loss induced by Aβ.

Line 28: “almost of DL intensity”, please reformulate

Answer: It was reformulate

Line 43: “presents”, please change to “present”

Answer : Thanks for your suggestion. It was corrected.

Lines 45/46: “ASTA might also prevent the intracellular and extracellular protein aggregates”, perhaps better: “… formation of…”

Answer : Thanks. It was corrected

Line 47: ” responsible of”, rather: “responsible for”

Answer: Thanks. It was corrected.

Line 57: “TG2 localization in the nucleus plays kinase activity”, please reformulate; localization cannot play activity; perhaps “TG2 localized… shows kinase activity”?

Answer: Thanks a lot for your suggestion. The sentence was changed.

Line 74: “[15,16] on”, rather: ” [15,16] of”

Answer: Thanks. It was corrected.

Line 113: “bought by”, rather: “bought from”

Answer: Thanks. It was corrected.

Line 131: Can the authors provide the permit number?

Answer: The permit number is 174/2017-PR. It was inserted in the text.

Line 193: omitted or subtracted?

Answer: The corrected word is omitted.

Line 237: “performed”, better “assay … was performed” or “viability… was evaluated”

Answer: Yes, we agree with your suggestion, therefore we changed the sentence.

Line 240: ” were choose”, should be “were chosen”

Answer: Thanks. It was corrected.

Line 247: is “re-establish” the right term? ASTA rather prevented loss of viability than re-established it (I do not believe in cell resurrection).

Answer: Thanks a lot for your suggestion. It was changed.

Figure 5. “Immunocytochemistry for anti-Nestin” is not an optimal legend. The authors studied nestin with antinestin. The same refers to Figure 6.

Answer: Yes, we removed “anti” in both figures (5 and 6)

Line 346: ”ASTA pre-treatment counteracted the oxidative status”, please re-formulate; perhaps “counteracted the modification of the oxidative status…”

Answer: Thanks a lot. The sentence was modified.

Figure 8A: Is a representative blot chosen for the Figure? It seems that the effect of DMSO was stronger on the blot than represented by the mean value on Figure 8B.

Answer: Yes, it is.

Line 523: “C-terminus of methionine Met-35”, not clear; “C-terminal methionine”?

Answer: Yes, it is. It was corrected. It was typing error.

Reviewer 2 Report

Astaxantin is a carotenoid differed from other representatives of this class of natural products such as <beta>carotene by the presence of oxidized functional groups on the both ends of the molecule that allows him to penetrate cell membranes and be active on the both sides of these membranes. This carotenoid is a natural product characteristic for marine microalgae including Haematococcus pluvialis. This substance has anti-oxidant, anti-tumor and anti-inflammatory, as well as immunomodulatory activities. It may be a potential therapeutic preparation for the cure of Alzheimer disease. The disease is linked with oxidative stress and accumulation of proteins, including amyloid-beta (A<beta>), a neurotoxic protein and a substrate of tissue transglutaminase (TG2). The authors showed the resistibility olfactory ensheathing cells exposed to A<beta>(1–42) or by A<beta>(25–35) or A<beta>(35–25), as well as decresase of the TG2 expression. The finding of all these effects and its evidence by different biochemical methods may be useful for understanding the mechanism of action of astaxantin.

 The article is well written and is interesting but I strictly recommend to present structural formula of astaxantin as a separate figure. The article may be published after very minor correction.

Author Response

Reviewer 2:

Astaxanthin is a carotenoid differed from other representatives of this class of natural products such as <beta>carotene by the presence of oxidized functional groups on the both ends of the molecule that allows him to penetrate cell membranes and be active on the both sides of these membranes. This carotenoid is a natural product characteristic for marine microalgae including Haematococcus pluvialis. This substance has anti-oxidant, anti-tumor and anti-inflammatory, as well as immunomodulatory activities. It may be a potential therapeutic preparation for the cure of Alzheimer disease. The disease is linked with oxidative stress and accumulation of proteins, including amyloid-beta (A<beta>), a neurotoxic protein and a substrate of tissue transglutaminase (TG2). The authors showed the resistibility olfactory ensheathing cells exposed to A<beta>(1–42) or by A<beta>(25–35) or A<beta>(35–25), as well as decresase of the TG2 expression. The finding of all these effects and its evidence by different biochemical methods may be useful for understanding the mechanism of action of astaxantin.

The article is well written and is interesting but I strictly recommend to present structural formula of astaxantin as a separate figure. The article may be published after very minor correction.

Answer: Thanks a lot for your suggestion. We added a supplementary figure in the manuscript.

Reviewer 3 Report

In the present study, the authors investigated the potential to reverse the toxic effect of Aβ peptides with astaxanthin in olfactory ensheathing cells. The article is interesting and suggests that astaxanthin could be used as a new AD therapeutic compound. However, there are the following issues that need to be addressed.

1. There are a few grammatical errors, e.g. lane 27 almost should be removed, lane 43 presents should be present, lane 64 does not make sense, lane 136 add should be added, lane 204 it should be added “used as secondary antibodies”, lane 205 it should be added “to visualize” etc.

2. Please explain in more detail the role of TG2 in AD

3. Please add more information on the DL. DL is increased when there is mitochondrial damage?

4. Please increase the letters or alter the font in Figure 1A.

Author Response

Reviewer 3:
In the present study, the authors investigated the potential to reverse the toxic effect of Aβ peptides with 
astaxanthin in olfactory ensheathing cells. The article is interesting and suggests that astaxanthin could 
be used as a new AD therapeutic compound. However, there are the following issues that need to be 
addressed.
1. There are a few grammatical errors, e.g. lane 27 almost should be removed, lane 43 presents should 
be present, lane 64 does not make sense, lane 136 add should be added, lane 204 it should be added 
“used as secondary antibodies”, lane 205 it should be added “to visualize” etc.
Answer: Thanks, your suggestions were performed, all sentences were corrected.
2. Please explain in more detail the role of TG2 in AD
Answer: Some details were added in the text. 
3. Please add more information on the DL. DL is increased when there is mitochondrial damage?
Answer: “the text after line 512 has been redrafted for clarity and a new reference has been added. 
The results obtained in the various cases cited do not allow us to give a definitive answer to your 
question but allow us to state that agents that target the mitochondria cause measurable changes in 
DL.
4. Please increase the letters or alter the font in Figure 1A.
Answer: it was corrected
